# The Water Safety Plan Approach: Application to Small Drinking-Water Systems—Case Studies in Salento (South Italy)

**DOI:** 10.3390/ijerph18084360

**Published:** 2021-04-20

**Authors:** Francesca Serio, Lucia Martella, Giovanni Imbriani, Adele Idolo, Francesco Bagordo, Antonella De Donno

**Affiliations:** Department of Biological and Environmental Sciences and Technology, University of Salento, via Monteroni 165, 73100 Lecce, Italy; lucia.mrt93@gmail.com (L.M.); giovanni.imbriani@unisalento.it (G.I.); adele.idolo@unisalento.it (A.I.); antonella.dedonno@unisalento.it (A.D.D.)

**Keywords:** environmental health, water quality, water safety plan, prevention, human health, environmental hygiene

## Abstract

Background: The quality of water for human consumption is an objective of fundamental importance for the defense of public health. Since the management of networks involves many problems of control and efficiency of distribution, the Water Safety Plan (WSP) was introduced to address these growing problems. Methods: WSP was applied to three companies in which the water resource assumes central importance: five water kiosks, a third-range vegetable processing company, and a residence and care institution. In drafting the plan, the terms and procedures designed and tested for the management of urban distribution systems were applied to safeguard the resource over time. Results: The case studies demonstrated the reliability of the application of the model even to small drinking-water systems, even though it involved a greater effort in analyzing the incoming water, the local intended use, and the possibilities for managing the containment of the dangers to which it is exposed. This approach demonstrates concrete effectiveness in identifying and mitigating the dangers of altering the quality of water. Conclusions: Thanks to the WSP applied to small drinking-water systems, we can move from management that is focused mainly on verifying the conformity of the finished product to the creation of a global risk assessment and management system that covers the entire water supply chain.

## 1. Introduction

In the last 100 years, water consumption has increased six-fold and continues to grow throughout the world [1] as food security, health, economic growth, and ecosystems all depend on water resources that are vulnerable to the impacts of global warming. In certain areas of the world, where population growth is more intense and climatic conditions are less favorable, despite being a renewable resource, water tends to become increasingly scarce. In addition, anthropic actions, which do not in themselves alter the total amount of water available but modify its quality, make it unusable from a sanitary point of view, contributing to worsening the depletion of water resources. In Italy and, in particular, Apulia (Southern Italy), the water emergency has experienced alternating phases of crisis and progressive worsening over the years [2,3]. Against this background of not-comforting prospects concerning the water availability of the territory, various initiatives to investigate the related problems and actions to define the planning necessary for containment measures have been launched [4,5,6]. The remediation actions set have followed a logic of “prevention”, with accurate targets for reducing loads in relation to specific needs and the intended use of each water body, corresponding from time to time to evaluations of the effects of the actions taken. In Apulia, the greatest value of all water availability is assumed by the fraction of drinking water, managed by the Apulian Aqueduct (AQP) through seven sources that are also located in the neighboring regions. The AQP, built in Puglia in 1906, conveys water from the spring of Sele River and Calore River in Campania to the most southeastern end of Italy. It has the biggest water supply network in Italy and Europe and supplies the drinking needs of 254 towns in Puglia. The water network has an extension of 20,000 km and serves more than 4,000,000 people [7]. The quality of AQP water is constantly analyzed, even in real-time, along its entire journey from the sources (springs, reservoirs, wells, and public fountains) to its arrival in the municipalities of Apulia and part of Campania. The water drawn from the water supplies is subjected to a purification treatment, according to the classification of raw water, carried out by the competent authorities, as required by the Environmental Code. The analysis is carried out in five laboratories located in Bari, Foggia, Lecce, Taranto, and Brindisi, which deal with analyzing distributed water and wastewater. Many laboratories are located at the surface water purification plants (Sinni, Locone, Pertusillo, Fortore, and Conza) to ensure that through the purification process, the natural raw water reaches a quality standard in line with the legal requirements. The purity and monitoring of the water are also guaranteed by additional disinfection stations and automated analysis units located in the main network nodes. The main feature of this adduction system is the strong degree of interconnection among different sources that allows the transfer of the water resource from one source to another, following changes in demand and compensating for the variable production rates of the different sources.

The management of such a large and interconnected network involves many problems of control and efficiency of distribution, for example, physical damage to the networks caused by climatic events, such as frost waves in the winter season, and depletion of many supply areas due to prolonged drought in the summer season.

Another important issue concerns the quality of water for human consumption, which is an objective of fundamental importance for the defense of public health. In fact, diseases due to the contamination of drinking water represent a risk to human health. For this reason, the current legislation on the protection of water for human use requires compliance with the minimum requirements of health and physical, chemical, microbiological, and radiological quality for drinking water.

Water quality is a determining factor for the suitability of a resource with respect to the requirements of use or species [8]. It can be influenced by climatic, geomorphological, geochemical, and biological factors inherent in the hydrological cycle, as well as by anthropogenic influences [8,9]. Access to safe drinking water and adequate sanitation and hygiene has the potential to prevent most water-borne diseases, including leptospirosis, cholera, and intestinal nematode infections [8]. At present, there are few water quality guidelines that address the full range of uses (e.g., domestic, agriculture, industry). In recent years, the International Water Resources Association (IWRA) has actively supported global initiatives related to water quality [10] and, in particular, domestic water use, which includes drinking water and water for domestic use, including food preparation, washing, and personal hygiene. In recent years, important research and guidelines related to the direct impact of the use of drinking water on human health have been provided on international and national levels [11]. This provides evidence of how domestic water quality can be effectively monitored and guided. Most countries have some form of routine regulation and monitoring of water quality, set at different scales, and provide different forms of guidance, such as indices, technical parameters, and implementation tools.

Water management quality is an important part of natural resource governance [12] that oscillates between different types of governance systems, ranging from protecting water sources, mitigating water pollution, maintaining and monitoring water quality standards, and implementing enforcement directives [13,14]. The local governance bodies are most vulnerable to water quality problems; they are also at the most proximate level to address these problems and are responsible for maintaining water quality [15,16]. Assurance of water quality, therefore, requires the formulation of a regulatory framework and institutional process.

In this regard, to ensure a high quality of water supplied to users and minimize the risk of drinking water, it is essential to base the management of water systems on risk analysis [17]. To address these growing problems in the drinking field and in order to strengthen the control system at the points of delivery deriving from the aqueduct network, used by different categories of consumers, the Water Safety Plan (WSP) was introduced. The goal of the Water Safety Plan is to integrate elements of site-specific analysis necessary to guarantee the healthiness of the water resource, taking into account any possible danger of microbiological, physical, chemical, or radiological nature that can occur within the system, evaluate the measures of the control systems in force, and propose new strategies aimed at reducing the risk value in order to fall within the regulatory limits.

In Italy, Legislative Decree n. 31/2001 [18], developed at the European level with European Directive 98/83/EC [19], defines three levels of water quality governance: the public authority, the managing body, and the user. Furthermore, the decree establishes the minimum quality requirements (sanitary and chemical, physical, microbiological) at the point where the water is available for human consumption and the responsibility for quality assurance. In particular, the decree establishes an integrated control of the quality of water intended for human consumption, including the private operator who is entrusted with the quality guarantee through routine control of the minimum requirements and the public authority entrusted with verification compliance with the quality criteria. The decree also establishes that the water network manager has responsibility up to the point of delivery (connection), after which the responsibility falls on all those who use that water for different purposes (e.g., the food industry, catering, bottling).

Recently, European Directive 98/83/EC [19] was modified with the new Directive (EU) 2015/1787 [20], implemented in Italy by the Ministerial Decree of 14 June 2017, which introduces the use of the WSP for water operators. The issuance of this decree represents a fundamental turning point in strengthening the quality of the water, thus passing from reactive control to proactive control.

The WSPs formulated by the WHO in 2004 [21], subsequently transposed to the regulatory level, therefore constitute an integrated prevention and control system based on site-specific risk analysis extended to the entire hydrodrinking chain, which marks a fundamental step to strengthening the quality of water to protect human health. In 2009, the WHO published a manual that describes the step-by-step WSP procedure [22]. Recently, the WSP has been included in European Directive 2015/1787 [20], which concerns the water quality intended for human consumption. Appropriate implementation of the WSP, therefore, offers an important opportunity to engage and promote preventive risk management within water services [23]. These reasons have prompted several nations to implement the WSP within their own territories. To date, WSPs are being implemented to varying degrees in 93 countries globally, with 30% of countries at an early adoption stage; 46 countries report having policy/regulatory instruments that promote or require WSPs, and, in another 23 countries, such instruments are under development [24], for example, in France [25], China [26], Germany [27], Portugal [28], India [29], and Italy [30].

The degree of WSP implementation and the impact on drinking-water quality varies significantly between European countries and with the development level of the water supply and the resources available. In all countries, there are reports of many benefits from WSP application, such as improved system management of water supplies; increased awareness, knowledge, and understanding among staff; improved communication and collaboration with other stakeholders, including within water supply companies; and improved water quality [31,32,33].

The aim of this paper is to apply the Water Safety Plan locally to three companies located in Salento, in the province of Lecce, such as water kiosks, a third-range vegetable processing company, and a sociorehabilitative structure. In drafting the plan, terms and procedures designed and tested up to now in the management of urban distribution systems have been applied, which are certainly equipped with a broader vision aimed at safeguarding the water resource over time.

## 2. Materials and Methods

The Water Safety Plan for this study was developed following the guidelines for risk assessment and management in the supply chain of drinking water [30], based on WHO recommendations. The WHO guidelines for drinking-water quality recommend the water safety plan (WSP) approach, a holistic risk assessment and risk management approach, to consistently ensure the quality of drinking water from catchment to consumer [24]. The key steps of a WSP are:description of the water system;system assessment, including the identification of hazards and hazardous events, as well as the assessment of risks;controlling risks by implementing control measures that prevent risk from manifesting, establishing operational monitoring to confirm the effectiveness of established control measures and defining corrective actions; andverification and auditing to confirm the safe operation of the whole system.

Water safety planning leads to a better understanding of the supply system, identifies potential sources of contamination, assesses risks to health, informs about technical and operational measures, and stipulates effective operational monitoring [24]. The WSP approach focuses on managing risks throughout all steps in the water supply chain, from source water catchment through treatment processes to storage, distribution, and handling of drinking water [24]. Thereby, it supports the achievement of health-based targets set by regulations and leads to stepwise improvement of the water supply system.

### 2.1. Study Area

The Water Safety Plan was applied to three companies in the Province of Lecce, where food is handled and the use of water is of central importance:five water kiosks that represent public supply points for natural and sparkling chilled water;a third-range vegetable processing company;a residence and care institution.

The availability of water in the examined companies is twofold and guaranteed by two sources of water supply: AQP and groundwater (artesian well).

AQP controls water quality parameters, and the only possibility of contamination is a possible release by water supply system elements; it is used as-is. The water that comes from the artesian well is treated by a chlorinator before being released into circulation, thanks to a presence of a storage tank in the company area.

### 2.2. Water Safety Plan Approach

The WSP model is developed through an iterative process, through several steps described in Figure 1.

#### 2.2.1. Establishing the Multidisciplinary Team

To ensure the realization of the plan, the first step is the establishment of a group of experts, with their responsibilities and authority defined within the organization, who have in-depth knowledge of the territory and its sources of water, the water treatment processes, and distribution networks.

In the experiment conducted, the group was made up of a food business operator, a technician responsible for the maintenance of the water network, and a technical–scientific referee for the analysis of the collected data.

#### 2.2.2. Water System Description

The description of the involved companies and the identification of the water supply system elements were carried out by analyzing the historical data of the plant and an on-site inspection and are represented by a flow chart, easily accessible by each member of the team. It must concern all the phases and operations carried out along the entire supply chain, the infrastructure and resources present or to be built and installed soon. In this work, three case studies of the application of a WSP in drinking-water supply systems in different environmental and cultural contexts of the province of Lecce (Salento, Italy) are described and analyzed.

#### 2.2.3. Water Supply System Assessment

Dangerous events can occur in any phase of the water supply system, and, for this reason, once the water system is estimated, the next step is to identify all the possible dangers and dangerous events in each phase of the drinking water system. These are hazards defined as physical, biological, chemical, or radiological agents that can cause damage to public health and hazardous events such as pollution, natural disaster, and accidental contamination.

The associated risk factors are identified according to the guidelines of the WHO Water Safety Plan [22] and Istisan Reports 14/21 [30], first of all, without taking into account the treatment facilities in the water supply chain, classifying them as “significant”, “uncertain”, or “not significant” on the basis of the impact they can have on human health.

Subsequently, a risk assessment is carried out again; the residual risk is classified as high (requires urgent improvement), medium (it is necessary to strengthen the control measures), or low (if it does not represent a need for intervention) depending on the effectiveness of each control measure present in the plant.

#### 2.2.4. Verification

The final check allows us to check the overall effectiveness of the WSP applied to the drinking water system, ensuring optimal levels of quality of the water supplied and protecting the health and safety of consumers. Sampling and laboratory analysis are conducted to verify that the controls are functioning accurately [34], including water sampling and microbiological checks to ensure water quality in all aspects of production processes. The choice to carry out only microbiological analyses was determined by the fact that the water used in most companies is supplied by the AQP, which is responsible for the potability of the water up to the connection point of the internal networks, where the only possibility of contamination is due to eventual release by supply system components.

The evaluation of drinking water parameters concerns the search for a specific class of bacteria, i.e., the indicator micro-organisms of fecal pollution, such as total coliforms, *Escherichia coli*, or enterococci. Another indicator parameter that allows us to define whether the water is of good quality is the total microbial count on agar at 37 and 22 °C.

#### 2.2.5. Review of the Risk System

The last phase includes the development and integration of an improvement program for the control of all dangers and associated risks. To maintain performance over time, operational monitoring is required, where an internal or external subject with the function of auditor continuously checks that each measurement gives compliant results [30].

## 3. Results

In this work, three case studies of the application of a WSP to drinking water supply systems in different environmental and cultural contexts of the province of Lecce (Salento, Italy) were described and analyzed. For each case study, the description of the water supply system, the adopted process for the implementation of the WSP, the obtained results, the critical issues, and the optimization proposals are reported.

### 3.1. Case Study No. 1: Water Kiosks

#### 3.1.1. Water System Description

The company carries out, at the operating units located in various municipalities in the province of Lecce, the business of selling chilled, sparkling, and natural drinking water. The structure of the kiosk consists of a compact body, on the sides of which are the supply compartments for customers. Access to the technical room is allowed only to authorized personnel in order to carry out maintenance and control of the equipment. All the equipment of the water system is made of material suitable for contact with drinking water, being washable and disinfectable. Furthermore, the kiosks are equipped with an automatic control system using a microprocessor capable of monitoring the operation of the various equipment, signaling maintenance needs, and programming the frequency and duration of the sanitization phases.

The inlet water is supplied by AQP and is, therefore, in compliance with Legislative Decree 31/2001 [18]; the plant (Figure 2) is able to deliver from 150 to 500 L/h and consists of:a water pressure reduction section, if more than 6 bar;three filters, the first of which is 50 µm by 11 “in diameter for the elimination of sand and foreign bodies, the second with a porosity of 1 µ by 20” in diameter, and the third is an activated carbon filter for the removal of dissolved organic pollutants such as chloramines and trihalomethanes;a 24 W UV lamp;a refrigeration section;a carbonation section;a 10 W UV lamp;dispensing spouts.

**Figure 2 ijerph-18-04360-f002:**
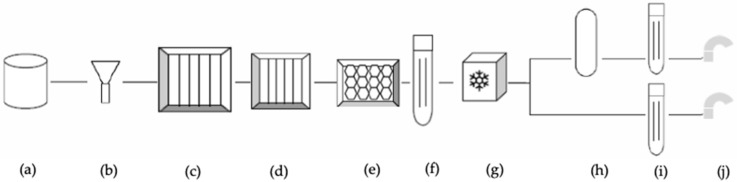
Process diagram of the water kiosks’ water supply chain: (**a**) water source—AQP, (**b**) pressure reduction system, (**c**–**e**) filters, (**f**) UV lamp, (**g**) cooler, (**h**) carbonation cylinder, (**i**) UV lamp, and (**j**) dispenser.

#### 3.1.2. Water Supply System Assessment

The risk assessment for the five kiosks located in different municipalities of the Province of Lecce is described in Table 1. Six hazards and dangerous events linked to the five plants were considered. The obtained results revealed a significant risk for the dispensing, disinfection, and maintenance phases. These risks are associated with accidental contamination (ineffective water disinfection due to inadequate removal of contaminants or malfunctioning of the UV lamp), with contamination from involuntary or deliberate actions (vandalism or soiling of the dispensing spouts), contamination due to incorrect hygiene management of the system, with a possibility of pollution of the system components during the maintenance phases, and, finally, exposure to values that are not in conformity with human consumption.

#### 3.1.3. Verification

As can be seen in Table 2, from the parameters sought, none of the carried-out checks recorded noncompliance, thus proving adequate the preventive measures undertaken.

#### 3.1.4. Review of the Risk System

Once the effectiveness of the control measures was validated, with appropriate microbiological analyses conducted on the five kiosks (Table 2), the residual risk was assessed (Table 1). In this specific case, the residual risk was low for both contamination due to deliberate and voluntary actions (thanks to the application of safety guards that surround the dispensing spouts) and the possible unsuitability of the water for human consumption (thanks to correct plant operation). The residual risks associated with accidental contamination and incorrect hygiene management are of a medium level.

The operational monitoring and control measures for the five kiosks are listed in Table 1. In order to ensure the safety of the final consumer over time and, therefore, the good management of the water system, the following actions were established:Schedule microbiological monitoring in particularly critical periods;Post instructions to the public;Expand the parameters sought to include chemical ones based on the technical data sheets of the products used in the maintenance phases.

### 3.2. Case Study 2: Third-Range Vegetable Processing Company

#### 3.2.1. Water Supply System Description

The company, which has been active for several decades, consists of a building with a ground floor, where the factory is located, and a first floor with administrative offices. The factory is divided into different areas: an entrance and raw material processing area, a transformation area (there are static and continuous cooking lines), a cooling area (in water), a blast chilling area, packaging areas, various storage areas (5 cells), offices, and a laboratory.

The structure is equipped with two distinct water distribution lines: one supplying water from the AQP (Figure 3), used at the level of the critical cooling points, and the other from the artesian well (Figure 4) falling within the company area, for all other operations.

As a disinfection system, both waters are chlorinated, but the artesian water supply is a specific type of free-flowing spring water that comes from underground wells. The waters, from chemical and microbiological points of view, may not be different, but they come to the earth’s surface a bit differently. In the water line (artesian water supply), no further investigations have been carried out as it is not used in the critical points of processing.

The control and maintenance activities are carried out by personnel trained for their execution and any emergency interventions.

The only storage point is represented by a groundwater collection tank, with a capacity of 40 m^3^; downstream of this, the water is treated by a chlorinator before being put into circulation in order to reduce any microbial charges.

#### 3.2.2. Water Supply System Assessment

The risk assessment was carried out on two water plants in the company, one coming from the AQP and one coming from an artesian well, and is represented in Table 3 and Table 4. As regards the water system supplied by the AQP, the results highlighted a high risk for the distribution and supply process and the maintenance phase. Three dangers and dangerous events were considered, of which accidental contamination (such as pollution by system components, with possible consequent bacterial colonization), contamination due to incorrect hygienic management (during the intervention and maintenance phases), and the unsuitability of the intended use. For the water line coming from an artesian well, four significant risks were identified, inherent to the phases of collection, treatment, supply, and maintenance. Six dangers and dangerous events were considered: accidental contamination (for possible pollution connected to the proximity of a state road, the presence in the aquifer of heavy metals, PAHs, pesticides, ineffective disinfection, pollution by the components of the system) and contamination due to incorrect hygienic management of the system.

#### 3.2.3. Verification

As shown in Table 5, from the parameters sought, abnormalities in the levels of coliform bacteria were found at a drinking water supply site.

#### 3.2.4. Residual Risk System Assessment

After assessing the presence of the control measures present, the system of residual risks was analyzed (Table 3 and Table 4). For the water supplied by the AQP, residual risk is high for accidental contamination and old parts of the system. There is medium residual risk related to incorrect hygienic management of the plant as the staff is properly trained on good practices and risk prevention, and there is low residual risk for the possible nonsuitability of water for human consumption, thanks to the drinking water parameters at the point of connection to the aqueduct. As for the water supplied by the artesian well, the residual risk associated with both accidental contamination and the intervention and maintenance phases is considered medium as the storage tank is suitably insulated.

#### 3.2.5. Review of the Risk System

The operational monitoring and control measures are listed in Table 3 and Table 4. In order to ensure the safety of the final consumer over time and, therefore, the good management of the water system, the following actions were established:the restoration of the continuity of internal surfaces, also for the pipes connected to equipment, end fittings, dispensers, and taps;the replacement of materials and substances not suitable for food contact;strengthening of the protection network;further chemical–physical and microbiological monitoring.

### 3.3. Case Study 3: Residence and Care Institution

#### 3.3.1. Water Supply System Description

The company is divided into two floors: a ground floor where the outpatient center (gyms, therapy rooms), kitchen, refectory, and administrative offices are located, and a first floor where the guest residences are located. The water supplied by the AQP is collected in a tank (storage necessary to overcome the pressure reduction in the summer period), and, from there, the internal network runs and is divided into five distribution lines. The control and maintenance activities are carried out by personnel trained for their execution and any emergency interventions.

The plant supply chain (Figure 5) includes:Connection section at the AQP delivery point;Storage tank;Treatment facilities (sand filter, chlorinator, mesh filter, softener, UV lamp);Thermal power plant and delivery points;Chilled drinking water dispenser, equipped with UV lamp.

**Figure 5 ijerph-18-04360-f005:**
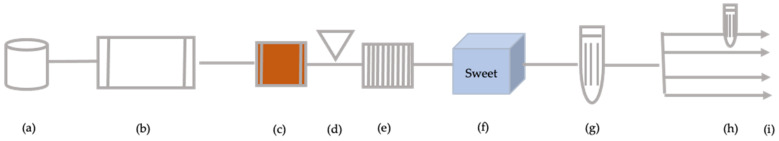
Process diagram of the residence and care institution: (**a**) water source—AQP, (**b**) water storage, (**c**) sand filter, (**d**) chlorination, (**e**) filter, (**f**) water softener, (**g**) UV lamp, (**h**) UV lamp, and (**i**) distribution networks.

#### 3.3.2. Water Supply System Assessment

The risk assessment carried out in the residential sociorehabilitation facility is outlined in Table 6. Taking into account the vulnerability of the subjects who are guests of the structure, the inspection conducted found a high risk in the delivery, maintenance, and storage phases. Four dangers and dangerous events were considered inherent: accidental contamination (possible pollution due to system components), contamination during maintenance phases, water conservation methods, and unsuitability for intended use (the compliance with drinking water parameters).

#### 3.3.3. Verification

As shown in Table 7, from the parameters sought, none of the checks carried out showed noncompliance, thus proving adequate the preventive measures undertaken.

#### 3.3.4. Residual Risk System Assessment

Once the effectiveness of the control measures was validated, the residual risk was assessed (Table 6). The residual risk inherent in the supply phase is high, as attention must be kept high due to the presence of vulnerable subjects; the residual risk related to water storage, on the other hand, is of medium risk due to the presence of downstream disinfection devices.

#### 3.3.5. Review of the Risk System

The operational monitoring and control measures are listed in Table 6. In order to ensure the safety of the final consumer over time and, therefore, the good management of the water system, the following actions were established:Monthly microbiological monitoring for the continuous evaluation of drinking water criteria;An intervention to isolate the tank entrance.

## 4. Discussion

### 4.1. Current Status of Water Usage and Water Pollution

The public water network of the Province of Lecce, managed by the AQP, develops from the Adriatic and Ionian branches that originate from the San Paolo reservoir, the last stretch of the Sele-Calore aqueduct. The aqueduct, which reaches all the municipalities of the Province of Lecce, satisfying the drinking water needs of the entire population, is also fed by groundwater drawn from 104 wells present throughout the province and concentrated mainly in the central and northern parts. The quality of the groundwater circulating in Salento is, therefore, decisive for human health as it directly impacts the quality of the water distributed by the public water network. Previous studies [34] have shown that the nitrate concentration in the groundwater of Salento exceeds the limits indicated in Legislative Decree 31/01. Nitrates are a chemical compound that is always present in water, with concentrations generally representative of the natural background and attributable to the contribution of precipitation and soil nitrification processes. Any increases in concentrations can be attributed to anthropogenic activities. Globally, in the last few decades, there has been a progressive increase in the concentration of nitrates in surface and groundwater in both agricultural areas and industrialized or urbanized areas, with increasing risks for aquifers destined for drinking purposes. Microbiological contamination has been known to exist for some time in the groundwater circulating in Salento [2,6] and is attributable both to the karst nature of the soil, which offers little resistance infiltration of any contaminants released on the surface, and the unsuitable disposal of sewage of civil origin, with particular reference to the widespread presence of leaky septic tanks. With regard to the issue of the supply of groundwater for drinking purposes, however, this risk can be reduced by purification treatments that are carried out on the water before it is introduced into the network. Finally, with regard to the indicator parameters, frequent exceedances of the limit indicated in Part C of Legislative Decree 31/01 have been recorded for salinity indices [35]. Rather than contamination of anthropogenic origin, this result is certainly correlated to the phenomena of marine water intrusion [3], which makes the underground waters of Salento particularly rich in mineral salts, with particular reference to chloride and sodium ions. In any case, as required by current legislation, the managing body has the task of constantly monitoring the quality of the water withdrawn from the supply sources and the water introduced into the water network, with particular attention to areas where particular situations have been highlighted, in order to guarantee the healthiness of the waters distributed through the public aqueduct.

### 4.2. New Methodological Approaches in the Management of the Quality of Water Intended for Human Consumption

Since 2004, the World Health Organization has recommended the adoption of its Water Safety Plan (WSP) globally to reduce the risk of contamination related to the use of water intended for human consumption and to ensure the protection of human health. The scientific basis on which the WSP approach is mainly focused is the analysis of the risks of contamination to which water can be exposed to in a drinking water supply system, from the capture to the point of distribution, with the primary objective of protecting human health. In this context, the WSP redefines the limits of the quality control systems of water intended for human consumption, using an innovative approach compared to the conventional sample monitoring of distributed water. The evolution of knowledge has highlighted the criticalities of the strategies solely focused on verifying the conformity of the finished product, shifting the interest towards a global risk assessment and management system extended to apply to the entire water supply chain. This system also makes it possible to manage the risk associated with the onset of extreme climatic events (such as floods and earthquakes) and/or the presence of emerging contaminants in water for human use because, during the drafting of the WSP, all the dangerous events that can contaminate the water, even the less frequent and rare ones, are taken into account [30]. The definition of a small purification system is not universal, but for each country, the competent authority provides a different description based on various factors, such as the number of people served, the complexity of the system, or the flow of water treated. In Italy, the definition of a small drinking-water system is reported in Legislative Decree 31/2001 [18] and is almost identical to that of the European community. However, the Italian legislation does not take advantage of the possibility left by European legislation to the Member States to exempt from the regulation the water from a single source that delivers, on average, less than 10 m^3^ per day or supplies less than 50 people. Therefore, in Italy, all water intended for human consumption is subject to the current legislation [18]. In Italy and Europe, no specific regulation has ever been issued regarding small drinking-water systems; therefore, the values indicated by the two regulations are assumed for both large and small drinking-water systems.

### 4.3. Application of the WSP to Case Studies: Advantages and Problems

The introduction of the WSP supports the identification of simple and convenient actions to be taken to protect and improve small water systems.

Therefore, it is important that the health authorities emphasize both the importance of safe drinking water distribution to local communities and the role and responsibilities of the operators of small drinking-water services for the health of the communities themselves. At the same time, it is important that local health authorities promote the process of implementing WSPs at the level of small drinking-water systems as an effective strategy to enhance some aspects in many critical cases. Our three case studies have revealed different scenarios that emerge through the collection of monitoring data. In Case Study 1, the equipment of a system suitable for the needs of water supply and cooling, only recently installed, shows all the advantages of updating the water treatment and disinfection techniques: on each of the five monitoring points, two checks, within eight months, of the microbiological parameters (coliform bacteria, *E. coli*, intestinal enterococci, viable microorganisms at 22 and 37 °C) were performed. None of the performed checks showed noncompliance, proving the adequacy of the preventive measures taken. In Case Study 2, a third-range vegetable processing company that has already been active for several years, the system shows signs of wear in the materials of the nonrenewed pipelines and the terminal parts connected to the equipment. In addition, this problem is compounded by those related to the coexistence of a drinking water line derived from the AQP and a water line from an artesian well located in the company area. Overall, of the 6 points analyzed, some anomalies were found: the first anomaly was found in the levels of coliform bacteria at a drinking water supply site, for which repeated monitoring was planned, and the cause of bacterial colonization in the elbow of a pipe was identified; the second anomaly was found when the micro-organism count values were exceeded in a well water supply point not intended for food contact. The coliform bacteria found do not represent a concern, as they are part of the static cooking line and, therefore, subjected to high temperatures at each processing cycle. In Case Study 3, the residence and care institution, the particular vulnerability of the exposed subjects led to more extensive monitoring. The structure, which has been in operation for decades, has undergone several maintenance and expansion interventions over time: at present, most of the water system has been renovated and replaced with more efficient materials and equipped with adequate treatment and disinfection devices. At the eight tested points, no nonconformities were found, but only high values for the viable micro-organisms at 22 and 37 °C in two sites, both supplied by the same branch of the network, the oldest. The carried-out work demonstrates the reliability of the application of the model even to small drinking-water management systems, even if work carried out on a smaller scale involves a greater effort in analyzing the characteristics of the plant, the incoming water, the local intended use, and the possibilities for managing the containment of the dangers to which the plant is exposed.

It is clear that the quality of water intended for human consumption can only be guaranteed if all elements of water quality governance (WQG) are respected, including:–Standardization of water quality through the monitoring of water quality and the reporting and assessment of the quality status;–Mitigation and protection, including maintaining water quality standards, protecting water sources, mitigating short- and long-term causes of water pollution, and risk assessment;–Application of WQG through the active involvement of local government authorities in the implementation of application directives, including the charging of penalties for water contamination and the issuing of licenses, permits and recommendations in the environmental impact assessment [12].

The application of the Water Safety Plan model to small water systems has demonstrated, on several levels, concrete effectiveness in identifying and mitigating the dangers of altering the quality of water. This precious resource is increasingly reduced by anthropogenic pressures (pollution, losses, waste); it requires protection and guarantees even within the risk prevention procedures already implemented in every facility connected to food activities. It becomes necessary that this line of thought is also adopted by large companies and nations, even before the obligation is established by the relevant legislation.

## 5. Conclusions

The Italian scene is characterized by a large number of small water managers who, in some cases, in addition to the structural problems typical of these systems, can be significantly affected by economic constraints and investment difficulties, as they have to integrate this responsibility with other important sectors of use (such as schools and social services) that the local authority is required to ensure, together with the water service. Furthermore, small water supplies, serving communities or individual households, should be able to rely on the selection of water sources of the best possible quality and on the protection of this quality through the use of multiple barriers (usually by means of protection systems at the capture point) and maintenance programs for operating systems. According to the WSP, control strategies in drinking water management systems are redefined, moving from an approach mainly focused on verifying the conformity of the finished product to the creation of a global risk assessment and management system that covers the entire water supply chain. The WSP substantially strengthens the system of controls at the withdrawal points on the AQP network, with the integration of site-specific analysis elements that guarantee a safe water supply, shifting interest towards the creation of a global risk assessment and management system extended to the entire water supply chain, from collection to the final user. The WSP’s goal is to drastically reduce the possibility of contamination of water intended for human consumption through water treatments that are properly designed, performed, and controlled and to prevent any recontamination during the storage and distribution of the water, up to the point of delivery. At the same time, it is essential to implement a continuous operational monitoring system on essential parameters, also with an early-warning function, in order to be able to react promptly to abnormal changes in water quality, which could lead to health and hygiene risks. The WSP has a preventive approach (risk assessment and management) and is no longer retrospective (downstream control), which intends to improve monitoring plans to rationalize and adapt processes in order to optimize the sanitation requirements of the distributed water. Previously, the frequency and sampling points were established by Monitoring Plans, but with the Ministerial Decree of 14 June 2017, the parameters and frequencies are now established on the basis of a risk assessment. The competent authorities will be able to decide which parameters to monitor on the basis of constant mapping and a careful analysis of the hydrographic and productive peculiarities of the territory and a serious risk assessment. The new methodologies, which allow the increase or decrease in the frequency of sampling in the supply areas and the introduction of new substances to be monitored in the event of specific events, will make it possible to abandon an analysis model that is currently very standardized and, in many ways, unsuitable for reacting or anticipating critical events that require a very fast and flexible response. This is work that will be able to acquire experiences that will contribute to drafting prevention action protocols on the subject of water and health, also providing elements of knowledge to be shared at the national and European levels. This approach has demonstrated effectiveness in identifying and mitigating the dangers of altering the quality of water. It represents a very good model of prevention and mitigation of the risks associated with a drinking water network, able to rationalize and systematize criteria and methods and improve the compliance of processes with hygienic–sanitary requirements. It is particularly recommended for use in areas such as the Salento, where the water resources for drinking purposes are particularly subject to considerable anthropogenic pressures or originate from high vulnerability territories. The limitation of this study is represented by the assessment of the application of the WSP to a small number (three) of small local companies that are very close to each other; therefore, it is difficult to evaluate its applicability to other types of activities. Its strength, however, was that it verified that a strategy designed for a larger scale could be adapted and used at more minor levels as well. However, further investigation is needed, taking into consideration other types of activities in different territories with different sanitation problems.

## Figures and Tables

**Figure 1 ijerph-18-04360-f001:**
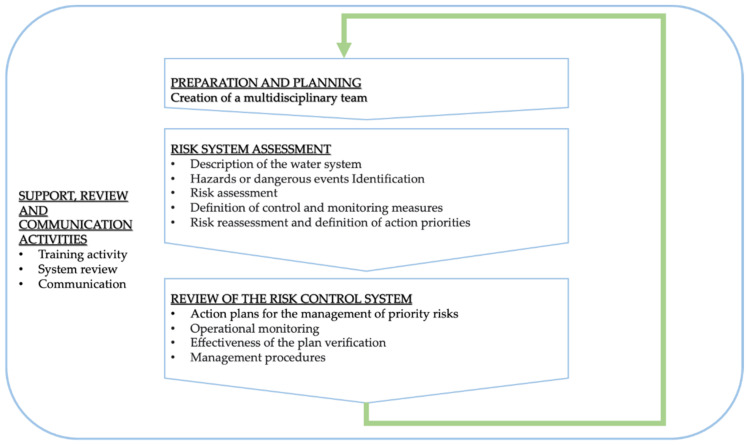
Methodological steps for developing a WSP to apply to a drinking-water system [30].

**Figure 3 ijerph-18-04360-f003:**
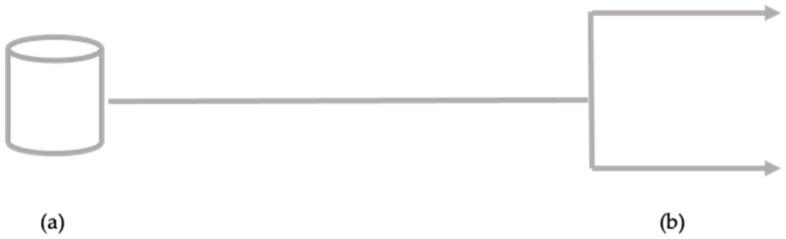
Process diagram of water pipeline from the AQP. (**a**) Water source—AQP; (**b**) distribution networks.

**Figure 4 ijerph-18-04360-f004:**
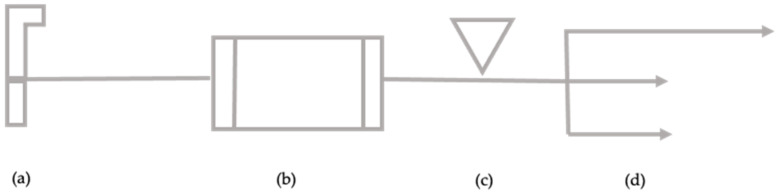
Process diagram of water pipeline from an artesian well. (**a**) Water source—artesian well, (**b**) water storage, (**c**) chlorination, and (**d**) distribution networks.

**Table 1 ijerph-18-04360-t001:** Identification of hazards and dangerous events: Case Study 1.

Water System Phase	Hazard and Hazardous Event	Associated Hazard	Hazard Types	Risk Rating	Control Measures	Residual Risk Rating	Improvement of Control Measures
Delivery	Contamination from external deliberate or involuntary actions	Vandalism and fouling of the spouts	Chemical Biological	Significant	Safety guard for dispensing points and daily inspection	Low	Public education billboard
	Water not suitable for the intended use	Exposure to values noncompliant with vulnerable groups of the population	Biological	Significant	Installation of two UV lamps	Low	Extension of the parameters sought
Disinfection	Accidental contamination	Ineffective disinfection due to UV lamp malfunction	Biological	Significant	Two UV lamps; interruption of the supply after a suspension of the electricity	Medium	Programmed microbiological monitoring in critical periods
Maintenance	Contaminations due to incorrect hygienic management of the system	Pollution in the intervention and maintenance phases	Chemical Biological Physical	Significant	Training on good practices for the disinfection treatment	Medium	Evaluation of organoleptic characteristics and chemical parameters
Treatment	Undersized treatments	Feed-in of untreated water	Biological	Uncertain	Inspection of the functionality of the system	Low	Not required
	Inadequate input water quality	Treatments not suitable	Chemical Biological Physical	Uncertain		Low	Not required

**Table 2 ijerph-18-04360-t002:** Results of the average calculation for each microbiological parameter on the 40 water samples * from the sampling point monitoring in Case Study 1. Limits and methods of analysis specified by Lgs.D. 31/01 [18] on the quality of water intended for human consumption. * The samples were collected in two seasonal sampling campaigns carried out in April and October on all points of the monitoring network. ** CFU: colony-forming unit.

	Microbiological Parameters				
Sampling Points	Coliform Bacteria (CFU **/100 mL) Mean ± St. Dv.	*E. coli* (CFU **/100 mL) Mean ± St. Dv.	Intestinal Enterococci (CFU **/100 mL) Mean ± St. Dv.	Viable Micro-Organisms at 22 °C (CFU **/100 mL) Mean ± St. Dv.	Viable Micro-Organisms at 37 °C (CFU **/100 mL) Mean ± St. Dv.
Kiosk 1	0.05 ± 0.07	0.01 ± 0.00	0.03 ± 0.05	21.27 ± 0.64	0.00 ± 0.00
	0.06 ± 0.05	0.15 ± 0.07	0.04 ± 0.05	0.05 ± 0.06	0.00 ± 0.00
Kiosk 2	0.00 ± 0.00	0.00 ± 0.00	0.00 ± 0.00	23 ± 0.05	12.17 ± 0.28
	0.00 ± 0.00	0.00 ± 0.00	0.00 ± 0.00	30.67 ± 0.51	7.90 ± 0.22
Kiosk 3	0.05 ± 0.07	0.01 ± 0.00	0.05 ± 0.06	4.87 ± 0.26	2.1 ± 0.22
	0.00 ± 0.00	0.00 ± 0.00	0.00 ± 0.00	0.00 ± 0.00	1.09 ± 0.13
Kiosk 4	0.00 ± 0.00	0.00 ± 0.00	0.00 ± 0.00	23.27 ± 0.47	14.94 ± 0.08
	0.05 ± 0.06	0.03 ± 0.05	0.05 ± 0.07	0.04 ± 0.05	8.31 ± 0.31
Kiosk 5	0.00 ± 0.00	0.00 ± 0.00	0.00 ± 0.00	47.01 ± 0.05	10.29 ± 0.28
	0.00 ± 0.00	0.00 ± 0.00	0.00 ± 0.00	19.83 ± 0.31	10.11 ± 0.61

Coliforms bacteria (ISO 930-1) are a large group of different types of bacteria that is commonly found in the environment, such as in soil and the intestines of animals, including humans. The main source of total coliforms in water is contamination from human and animal waste. A specific subgroup of this collection is fecal coliform bacteria, in which the most common member is *Escherichia coli* (ISO 930-1), considered the best indicator of fecal pollution and the possible presence of pathogens. Intestinal enterococci (ISO 7899-2) are bacteria that can be used as a marker to indicate fecal contamination of potable water. Their abundance in human and animal feces has led to their widespread use as a tool for assessing water quality worldwide. Viable micro-organisms at 22 °C and viable micro-organisms at 37 °C (prEN ISO 6222).

**Table 3 ijerph-18-04360-t003:** Identification of hazards and dangerous events: Case Study 2—AQP pipeline.

Water System Phase	Hazard and Hazardous Event	Associated Hazard	Hazard Types	Risk Rating	Control Measures	Residual Risk Rating	Improvement of Control Measures
Delivery	Accidental contamination	Pollution of system components	Biological	Significant	Replacement of old pipes with multilayer pipes	High	Restoration of fittings, dispensers, and taps
	Water not suitable for the intended use	Exposure to values noncompliant to vulnerable groups of the population	Biological	Significant	System designed exclusively for drinking water	Low	Extension of the parameters sought
Maintenance	Incorrect hygienic management of the system	Pollution in the intervention and maintenance phases	Chemical Biological Physical	Significant	Training on good practices for disinfection treatment	Medium	Use of materials and substances suitable for food contact

**Table 4 ijerph-18-04360-t004:** Identification of hazards and dangerous events: Case Study 2—Artesian well line.

Water System Phase	Hazard and Hazardous Event	Associated Hazard	Hazard Types	Risk Rating	Control Measures	Residual Risk Rating	Improvement of Control Measures
Water intake Treatment	Accidental contamination	Pollution by highway	Physical	Significant	Well insulation	Medium	Strengthening of the well protection network
		Presence of pesticides, heavy metals, PAHs	Chemical				Chemical, physical, and microbiological monitoring
		Ineffective disinfection of water	Biological		System design based on fluctuation data and supply interruption after suspension of the electricity		Chemical, physical, and microbiological monitoring Monthly check of free chlorine levels
		Pollution due to system components	Chemical Biological		None		Scheduling of line replacement interventions
Maintenance	Incorrect hygienic management of the system	Pollution in the intervention and maintenance phases	Chemical Biological Physical	Significant	Training on good practices for disinfection treatment	Medium	Evaluation of the organoleptic characteristics and of the chemical parameters

**Table 5 ijerph-18-04360-t005:** Results of the average calculation for each microbiological parameter of the 48 water samples * from the sampling point monitoring in Case Study 2. Limits and methods of analysis specified by Lgs.D. 31/01 [18] on the quality of water intended for human consumption. * The samples were collected in two seasonal sampling campaigns carried out in April and October on all points of the monitoring network. ** CFU: colony-forming unit.

Sampling Points *	Microbiological Parameters				
	Coliform Bacteria (CFU **/100 mL) Mean ± St. Dv.	*E. coli* (CFU **/100 mL) Mean ± St. Dv.	Intestinal Enterococci (CFU **/100 mL) Mean ± St. Dv.	Viable Micro-Organisms at 22 °C (CFU **/100 mL) Mean ± St. Dv.	Viable Micro-Organisms at 37 °C (CFU **/100 mL) Mean ± St. Dv.
Static cooking AQP line	1.21 ± 0.30	0.00 ± 0.00	0.00 ± 0.00	14.96 ± 0.13	0.01 ± 0.02
	0.00 ± 0.00	0.00 ± 0.00	0.00 ± 0.00	0.00 ± 0.00	0.00 ± 0.00
	4.18 ± 0.33	0.05 ± 0.00	0.04 ± 0,05	19.31 ± 0.51	15.6 ± 0.66
Lower dynamic cooling cooking AQP line	0.00 ± 0.00	0.03 ± 0.05	0.01 ± 0.00	16.92 ± 0.90	4.20 ± 0.31
Superior dynamic cooling cooking AQP line	0.05 ± 0.07	0.01 ± 0,00	0.05 ± 0.06	20.8 ± 0.63	1.2 ± 0.20
1st well packaging area	0.00 ± 0.00	0.00 ± 0.00	0.00 ± 0.00	0.03 ± 0.04	0.98 ± 0.40
2nd well packaging area	0.01 ± 0.00	0.05 ± 0.06	0.04 ± 0.05	172.52 ± 0.82	126.34 ± 0.57
Well raw material washing area	0.00 ± 0.00	0.05 ± 0.06	0.03 ± 0.05	2.96 ± 0.11	0.00 ± 0.00
	0.00 ± 0.00	0.00 ± 0.00	0.00 ± 0.00	0.00 ± 0.00	0.00 ± 0.00

Coliforms bacteria (ISO 930-1) are a large group of different types of bacteria that is commonly found in the environment, such as in soil and the intestines of animals, including humans. The main source of total coliforms in water is contamination from human and animal waste. A specific subgroup of this collection is fecal coliform bacteria, the most common member being *Escherichia coli* (ISO 930-1), considered the best indicator of fecal pollution and the possible presence of pathogens. Intestinal enterococci (ISO 7899-2) are bacteria that can be used as a marker to indicate fecal contamination of potable water. Their abundance in human and animal feces has led to their widespread use as a tool for assessing water quality worldwide. Viable micro-organisms at 22 °C, and viable micro-organisms at 37 °C (prEN ISO 6222).

**Table 6 ijerph-18-04360-t006:** Identification of hazards and dangerous events: Case Study 3.

Water System Phase	Hazard and Hazardous Event	Associated Hazard	Hazard Types	Risk Rating	Control Measures	Residual Risk Rating	Improvement of Control Measures
**Delivery**	Water not suitable for the intended use	Exposure to values noncompliant with human consumption of vulnerable groups of the population	Biological	Significant	Installation of disinfection devices Restoration of 50% of the pipelines Training of technical personnel	High	Monthly microbiological monitoring
	Accidental contamination	Pollution due to system components	Biological	Significant	Uniformity and sedimentation implemented in the tank 50% replacement of pipes Filters, chlorinator, softener, and UV lamp	Medium	Monthly microbiological monitoring and preordered maintenance of the plant segments
**Maintenance**	Incorrect hygienic management of the system	Pollution in the intervention and maintenance phases	Chemical Biological Physical	Significant	Training on good practices	Low	Monthly microbiological monitoring
**Storage**	Water conservation	Pollution caused by stagnation or infiltration	Biological	Significant	Disinfection devices	Medium	Insulation of tank spout Monthly microbiological monitoring

**Table 7 ijerph-18-04360-t007:** Results of the average calculation for each microbiological parameter on the 48 water samples * from the sampling point monitoring in Case Study 3. Limits and methods of analysis specified by Lgs.D. 31/01 [18] on the quality of water intended for human consumption. * The samples were collected in two seasonal sampling campaigns carried out in April and October on all points of the monitoring network. ** CFU: colony-forming unit.

Sampling Points *	Microbiological Parameters				
	Coliform Bacteria (CFU **/100 mL) Mean ± St. Dv.	*E. coli* (CFU **/100 mL) Mean ± St. Dv.	Intestinal Enterococci (CFU **/100 mL) Mean ± St. Dv.	Viable Micro-Organisms at 22 °C (CFU **/100 mL) Mean ± St. Dv.	Viable Micro-Organisms at 37 °C (CFU **/100 mL) Mean ± St. Dv.
Static cooking AQP line	1.01 ± 0.07	0.00 ± 0.00	0.00 ± 0.00	15.12 ± 0.17	0.15 ± 0.07
	0.00 ± 0.00	0.00 ± 0.00	0.00 ± 0.00	0.00 ± 0.00	0.00 ± 0.00
	4	0	0	19.29 ± 0.52	16.06 ± 0.89
Lower dynamic cooling cooking AQP line	0.01 ± 0.00	0.03 ± 0.05	0.01 ± 0.00	16.70 ± 0.57	3.98 ± 0.09
Superior dynamic cooling cooking AQP line	0.07 ± 0.05	0.03 ± 0.04	0.01 ± 0.02	21.02 ± 0.05	1.01 ± 0.06
1st well packaging area	0.00 ± 0.00	0.00 ± 0.00	0.00 ± 0.00	0.00 ± 0.00	1.02 ± 0.06
2nd well packaging area	0.05 ± 0.06	0.03 ± 0.05	0.05 ± 0.07	172.84 ± 0.33	126.34 ± 0.56
Well raw material washing area	0.01 ± 0.00	0.00 ± 0.00	0.01 ± 0.02	3.31 ± 1.08	0.01 ± 0.04
	0.00 ± 0.00	0.00 ± 0.00	0.00 ± 0.00	0.00 ± 0.00	0.00 ± 0.00

Coliforms bacteria (ISO 930-1) are a large group of different types of bacteria that is commonly found in the environment, such as in soil and the intestines of animals, including humans. The main source of total coliforms in water is contamination from human and animal waste. A specific subgroup of this collection is fecal coliform bacteria, the most common member being *Escherichia coli* (ISO 930-1), considered the best indicator of fecal pollution and the possible presence of pathogens. Intestinal enterococci (ISO 7899-2) are bacteria that can be used as a marker to indicate fecal contamination of potable water. Their abundance in human and animal feces has led to their widespread use as a tool for assessing water quality worldwide. Viable micro-organisms at 22 °C and viable micro-organisms at 37 °C (prEN ISO 6222).

## Data Availability

Data is contained within the article.

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
