# Peer review of "The Water Safety Plan Approach: Application to Small Drinking-Water Systems—Case Studies in Salento (South Italy)"

_ijerph, 2021, doi:10.3390/ijerph18084360_

Round 1
Reviewer 1 Report
PDF File attached

Reviewer 2 Report
Thanks for your interesting paper on the water safety plan.
The abstract is not well written. The way you wrote, methods, the conclusion in an abstract is not appropriate.
It is important to study the Global Compendium on Water Quality Guidelines from International Water Resource Association
This will gove your clear idea about the different aspect of the quality guidelines.
When reading the paper, the lacuna of your paper is that there is a lack of theoretical understanding of water quality governance/management issues. These sections need to be revised with more deepen study on water quality governance. Because water quality and water supply as demand, supply, and quality status issues, you have to look at the different aspects of the water quality governance at the local level.
Please read the theoretical analytical tool of water quality governance in the following study. You can catalyze the thread issue/potentiality and weaknesses of each step in the management process. This is will give more meaning to the technical clarification.
Withanachchi, S.S.; Ghambashidze, G.; Kunchulia, I.; Urushadze, T.; Ploeger, A. A Paradigm Shift in Water Quality Governance in a Transitional Context: A Critical Study about the Empowerment of Local Governance in Georgia. Water 2018, 10, 98. https://doi.org/10.3390/w10020098
Some additional readings are
- Wardropper, C.B.; Chang, C.; Rissman, A.R. Fragmented water quality governance: Constraints to spatial targeting for nutrient reduction in a Midwestern USA watershed. Landsc. Urban Plan. 2015, 137, 64–75. [Google Scholar] [CrossRef]
- Kayser, G.L.; Amjad, U.; Dalcanale, F.; Bartram, J.; Bentley, M.E. Drinking water quality governance: A comparative case study of Brazil, Ecuador, and Malawi. Environ. Sci. Policy 2015, 48, 186–195. [Google Scholar] [CrossRef] [PubMed]
- Moore, M.-L. Perspectives of complexity in water governance: Local experiences of global trends. Water Altern. 2013, 6, 487–505. [Google Scholar]
- Roussary, A. The reorganisation of drinking water quality governance in France. Responsibility- based governance and objective-driven policy setting in question. Rev. Agric. Environ. Stud. 2014, 95, 203–226. [Google Scholar] [CrossRef]
- Wuijts, S., Driessen, P. P., & Van Rijswick, H. F. (2018). Towards more effective water quality governance: A review of social-economic, legal and ecological perspectives and their interactions. Sustainability, 10(4), 914.
Your tables are a bit wordy. This may disturb the readers. My proposal is
For example
Table 6. Identification of hazards and dangerous events: case study 3 - column Improvement of control measures: please get only keywords
Table 2 is unclear.
The way the result presented could be more detailed. The current structure is more of a technical report.
Discussion section
You need to rewrite that discussion section with subtitles. When you have the theoretical framing with the above readings, you will able to have meaningful sub-titles for 3 to 4 sections. These need to be related to result in section.
Reviewer 3 Report
The article is very interesting and devoted to very important aspects related to water safety. The risk assessment is done fairly.
My critical remarks are as follows:
1. The Si units should be used in scientific papers,
2. All the pictures are very mean. Drawings should be corrected to meet the requirements of the flowchart of elements of water supply systems. All elements of water system should be labeled with number in the picture, and identified in the discription.
3. The English language should be improved in terms of water treatmet facilities and water supply systems elements.
4. In my assessment, the conclusions should relate to the three analyzed cases, pointing to some general rules for making decisions regarding water safety and pointing to differences in the approach in individual cases that are so different from each other.
Summing up, the article should be considered important and appropriate to read
Round 2
Reviewer 2 Report
Authors did the considerable development on the manuscript. However, the still authors do not correctly understand the comment on water quality governance elements and how they integrate into different steps in the water quality supply system at the local context in Italy. Please read the
You can develop your discussion section with the WQG elements into three main groups: - standardization, mitigation and protection, and enforcement (Withanachchi et al., 2018).
In conclusion, you need to mention the limitations and future further research possibilities.
